# The Power of Human–Algorithm Collaboration in Solving Combinatorial Optimization Problems

**Tapani Toivonen * and Markku Tukiainen**

School of Computing, University of Eastern Finland, 80110 Joensuu, Finland; markku.tukiainen@uef.fi
* Correspondence: tapani.toivonen@uef.fi

**Abstract:** Many combinatorial optimization problems are often considered intractable to solve exactly or by approximation. An example of such a problem is *maximum clique*, which—under standard assumptions in complexity theory—cannot be solved in sub-exponential time or be approximated within the polynomial factor efficiently. However, we show that if a *polynomial time* algorithm can query informative Gaussian priors from an expert $poly(n)$ times, then a class of combinatorial optimization problems can be solved efficiently up to a multiplicative factor $\epsilon$, where $\epsilon$ is arbitrary constant. In this paper, we present proof of our claims and show numerical results to support them. Our methods can cast new light on how to approach optimization problems in domains where even the approximation of the problem is not feasible. Furthermore, the results can help researchers to understand the structures of these problems (or whether these problems have any structure at all!). While the proposed methods can be used to approximate combinatorial problems in NPO, we note that the scope of the problems solvable might well include problems that are provable intractable (problems in EXPTIME).

**Keywords:** combinatorial optimization; bayesian optimization; NP-hardness; human-algorithm collaboration

## 1. Introduction

Human-in-the-loop (HITL) AI has gained quite a lot of attention during the past few years [1]. It has emerged to compete with autonomous systems in fields where the interference of a human user is required to solve certain problems that might be intractable for an autonomous AI. More specifically, HITL AI aims to find solutions to problems by involving the human learning process of an AI model. Such interaction can include, for instance, model adjusting, hyperparameter tuning, or data processing. As a new competitor of autonomous AI, human–algorithm collaboration has shown promising results in educational data mining and learning analytics [2]. Other promising fields of human–algorithm collaboration include human-in-the-loop optimization [3] and human–robot interaction [4].

Solving hard combinatorial optimization problems such as *maximum satisfiability* [5], *maximum clique* [6] or *minimum vertex cover* [7] is usually considered intractable. Consensus in computational complexity theory is that these problems cannot have efficient algorithms that would always yield correct results [8]. That is, no polynomial time algorithms to solve them exist. This further implies that some problems, such as finding the maximum-sized cliques or independent sets in graphs, cannot even be approximated well and efficiently at the same time [9], if widely assumed super polynomial lower bounds for the problems hold.

Solving such hard problems in general has not gained much attention in human–algorithm collaboration theory as the main focus has been on AI applications [10] such as learning [11]. In this paper, we show that, through human–algorithm collaboration, a combinatorial minimization or maximization problem that has a combinatorial complexity of $O(2^n)$ (or as a matter of fact, any $O(poly(n)^{poly(n)})$, can be solved efficiently up to an arbitrary multiplicative factor $\epsilon$, *if* the algorithm can query *Gaussian priors* from a human expert

during the execution. Combinatorial complexity means the size of the problem's search space. For instance, in the maximum clique, the combinatorial complexity is $O(2^n)$ but in the travelling salesman problem, the combinatorial complexity is $O(n!)$. Note that *any* NP optimization problem can be reduced into, say, a clique problem, which *has* a combinatorial complexity of $O(2^n)$. Further, we show that the scope of possible problems solvable with the proposed method includes even problems with super-exponential search spaces.

Bayesian optimization is a framework to solve hard function problems and is based on Bayesian statistics. Usually, before the actual Bayesian optimization, an *expert* provides the algorithm with some information about the problem that is being optimized in a form of a *Gaussian prior*. It reflects one's understanding on shape and smoothness of the objective function. Querying Gaussian priors from human experts is a very common assumption in Bayesian optimization literature [12]. In the *fully* Bayesian approach to AI and global optimization, the optimization procedure based on Gaussian processes expects the Gaussian prior $G(\mu, \Sigma)$ as part the input [13]. That is, the prior is decided by a human expert. Additionally, in fully Bayesian cases, the prior is expected to be correct. More specifically, the function being optimized is a realization of the previously given Gaussian prior (in Figure 1, 3 realizations of Wiener processes, for instance). In this paper, we make similar assumptions. We expect that the priors are *consistent* and *informative*. These assumptions are quire realistic and very standard, because the expert has access to previous function evaluations, problem instances and multiple prior sampling heuristics such as the maximum likelihood estimation.

Nevertheless, we stress that we do not claim to solve any notoriously hard problem in theoretical computer science. Instead, we show that human–algorithm collaboration can help us solve even those problems considered intractable in the future as this field matures.

This paper is organized as follows. First, we introduce the concept of Bayesian optimization. Second, we show how to reduce a class of combinatorial optimization problem instances to a *univariate* finite domain function, which can then be approximated by our human–algorithm collaboration Bayesian optimization algorithm that we introduce thereafter. Finally, we conclude our research and mention some future directions.

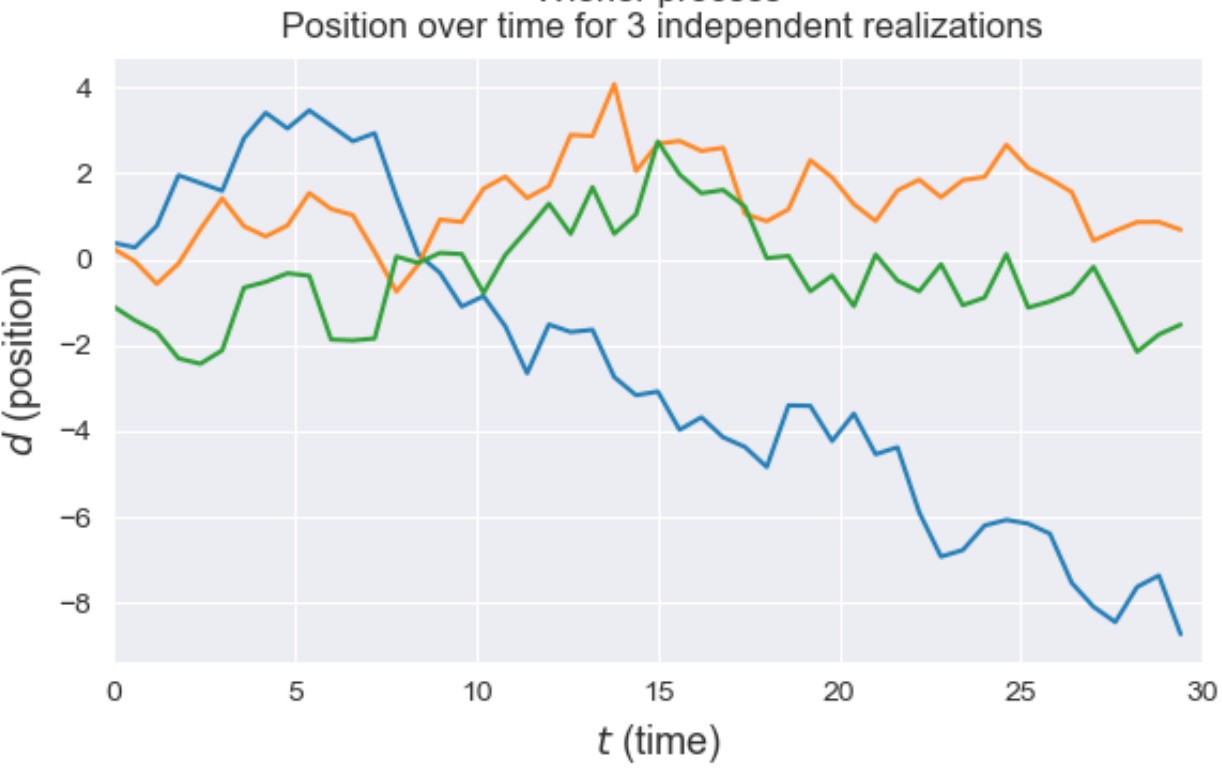

**Figure 1.** 3 realizations of the Wiener process.

## 2. Bayesian Optimization

Global optimization (GO) aims to find optimal value(s) of functions, called *objective functions*, either in finite or bounded domains [14]. The optimal values, depending on the context, can be the global maximal or minimal values. In general, global optimization is intractable—the number of function evaluations increases exponentially in the problem dimensions and exponentially in the domain size [15]. The GO problems for continuous functions and functions with finite domains, respectively are as follows:

$$\text{maximize } F(x), \ s.t. \ x \in [0,1]^d, d \in \mathbb{N} \tag{1}$$

$$\text{maximize } F(x), \ s.t. \ x \in N, N \in \mathbb{N} \tag{2}$$

In some cases, there can be more constraints besides the domain. Those problems, however, are not in the scope of this research. We also note that any minimization problem can be reduced into a maximization problem by simply using $F := -F$.

Bayesian optimization (BO) is a technique used in GO to search for globally optimal values in continuous, combinatorial, and discrete domains [16]. BO has a wide range of applications and recent years, its usage in optimizing hard black box functions has increased [17]. BO is often used in low data regimes where evaluation of the objective function is costly or not otherwise efficient [18]. These problems include, robot navigation and planning [19], tuning the hyperparameters of deep learning models [20], predicting earthquake intensities [21], finding optimal stock values in stock markets [22], and much more [23].

The advantage of the BO is that it can optimize any set of black box functions. That is, if one can only access function values and not, say, gradient information, then BO can be very efficient [17]. In addition, BO does not usually make any assumptions on the function it optimizes unlike multiple state-of-the-art optimization algorithms [16] that expect the objective function to be Lipschitz continuous [24], unimodal [25], or have a certain shape close to the global optimizer [26]. *Fully* BO, however, assumes that the user has some knowledge or expectation on the shape of the objective function, which is realized as a (Gaussian) *prior*. Moreover, the main difference between BO and traditional GO is that in BO, one is not expected to provide, say, a differentiable function—one just has to know what the kind of function it is to some extent.

BO is mainly based on Gaussian processes [27], which are stochastic process such that every finite collection of random variables has a multivariate normal distribution. It is completely defined by its covariance function $\Sigma$ (or covariance matrix in finite domains) and its mean function $\mu$ [16]: that is, $N(\mu, \Sigma)$.

In this paper, it is assumed that for *any* function $F$ is considered as

$$F \sim N(\mu, \Sigma) \tag{3}$$

and $\mu$ is some constant, say 0 for a centered Gaussian process. This is a very common assumption made in BO literature.

In BO, one places a (Gaussian) *prior* on the unknown and possible non-convex function. This reflects one's understanding of the function—whether the function is continuous, differentiable, smooth, unimodal, and so forth [16]. It is usually assumed that $\mu$ is 0 and hence, the prior is defined only by its covariance function (or covariance matrix) $\Sigma$. After the prior is set over the unknown function, an algorithm suggests points where the optimal values would lie. Based on these suggestions and function evaluations, the algorithm updates the prior to *posterior* and uses the posterior to suggest even better points where the possible optimum would be located. In many settings, the BO finds the global optimum much faster than, say, a brute force search [17]. The suggestions and how they are derived from the prior and posterior are defined by an acquisition function [16], for which multiple different possibilities exist. These include upper confidence bound [28], expected improvement [29], and Thompson sampling [30], to name few. A visualization of

BO with posterior distribution of functions, sample points, and an acquisition function can be seen in Figure 2.

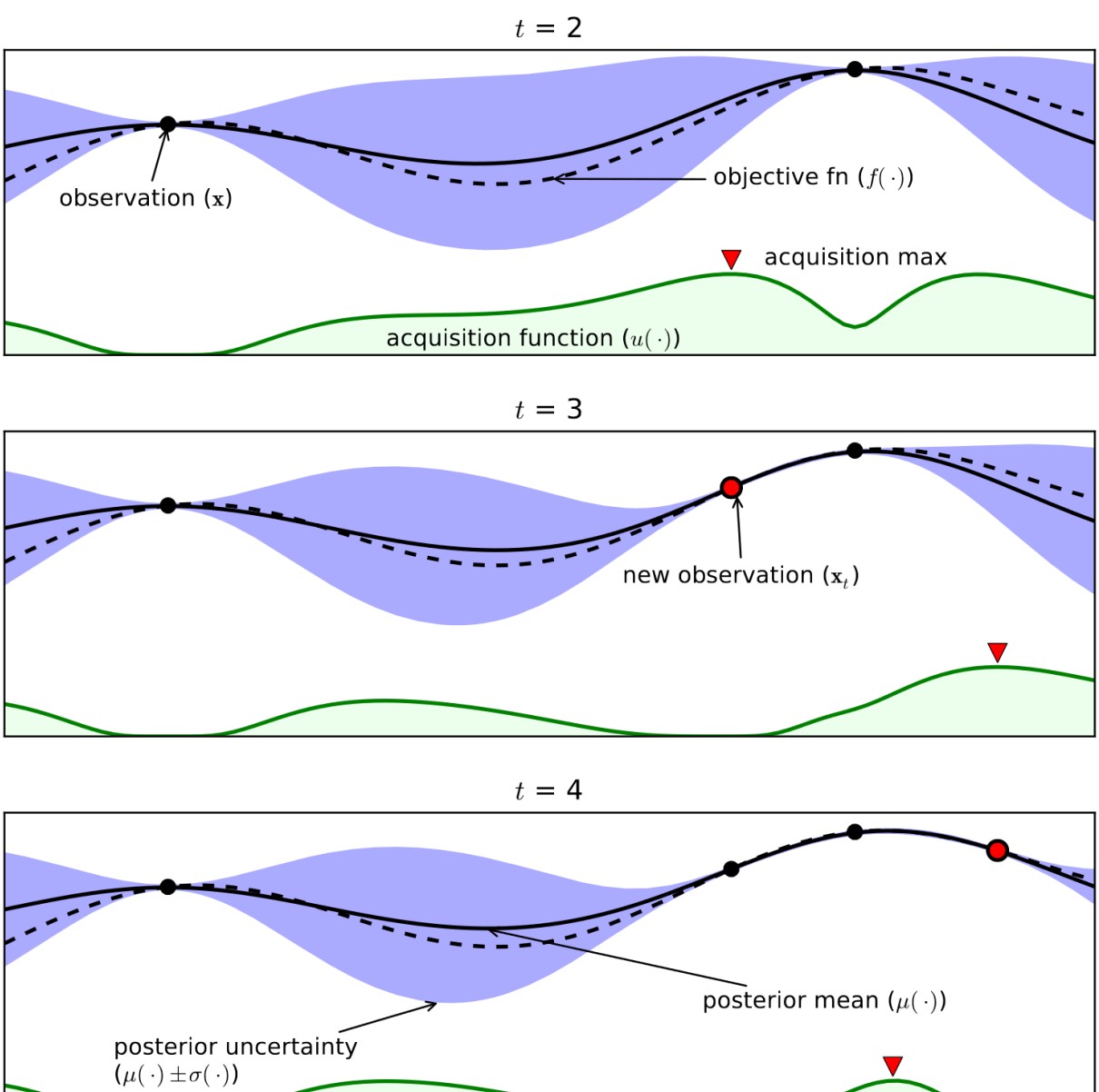

**Figure 2.** BO in 3 steps. The dotted function is the objective function to be optimized, the black function is the posterior mean, and the purple area is the confidence level of the acquisition function, i.e., UCB, EI, and so forth [31].

Multiple studies have shown that BO converges on the global optimum in reasonable time as shown in [16,17]. Reference [23] showed an convergence rate of BO to the expected global optimum (that is, the expected highest or lowest value of the function depending on the context) if the function near to global the optimum had a certain shape. Reference [32] showed similar results to a more restricted version of BO where the function was a realization of a Gaussian process called Brownian motion (also known as the Wiener process).

While it is tempting to praise BO for its ability to efficiently find optimal values of the structures, one of the downsides of the approach is that the locally and globally optimal values found are based on how well the prior of the Gaussian process is defined. If the black box function is a realization of the prior, then BO works surprisingly well. On the other hand, a poor choice of a prior can cause the algorithm to not converge at all. In BO

literature [16,17], it is usually assumed that an expert, who has a domain knowledge in the context of the objective function, provides a prior from which the objective function is realized.

The *regret* [18], which is used to give an idea of the convergence rate of a BO algorithm, is based on the expectations—that is, ratio between the *expected* global optimum and *expected* best value found in the function. The regret is defined as

$$r_T = \mathbf{E}[F_{sup}] - \mathbf{E}[Y] \tag{4}$$

where $F_{sup}$ is the global optimum and $Y$ is the best value found by a BO optimization algorithm. The expectation in the regret means that on average among all functions, the BO algorithm will return a value that is $r_T$, which is close to an expected optimum. The expected values are bounded by the prior and posterior distributions of the Gaussian process. In [27], the regret was defined as a multiplicative factor of the expected best value found and global optima

$$r_T := normregret = \frac{\mathbf{E}[F_{sup}] - \mathbf{E}[Y]}{\mathbf{E}[F_{sup}]} \tag{5}$$

which is closely related to the approximation ratio in approximation. In this paper, we complement the previous results of [27] to obtain the (super-)exponential convergence rate for BO. We restrict ourselves to the objective functions that can be *changed* while preserving the relative scale of the function values in the original objective function. We show that any combinatorial optimization problem with a combinatorial complexity of $O(poly(n)^{poly(n)})$ can be reduced to such a function. Later, we show how the (super-)exponential convergence rate can be derived using our human–algorithm collaboration procedure.

### 3. Combinatorial Problems as Univariate Finite Domain Functions

In this section, we show that any problem with a *combinatorial* complexity of $O(2^n)$ can be reduced to a *univariate* function with a domain of size $\Omega(2^n)$. The reduction is quite general and can also be applied to problems with a combinatorial complexity of $O(poly(n)^{poly(n)})$. Here, the different branching factor $m$ has to be considered.

The reduction algorithm (seen in Algorithm 1) assumes that the problem instance can be viewed as a decision tree (see Figure 3), where a left child arc of a decision node implies that the decision variable at the node is assigned as 0 and the right child arc implies the assignment of 1 (or vice versa). The function that the reduction produces, takes value in a domain $D = [D_0, D_1]$ and uses the value to find an assignment for the combinatorial problem instance. Finally, the function evaluates the combinatorial problem with the assignment derived from the input value.

The algorithm starts by ordering the decision variables randomly. Next, the algorithm creates an objective function that accepts a value from a finite domain $D = [D_0, D_1]$. Based on the value passed to the function, the function always divides the remaining domain into two equally sized halves and selects the half that contains the value. At each selection of the half, the assignment is updated with the corresponding decision variable value based on which the halves were selected. When the domain has been completely split so that no more intervals can be selected, the original combinatorial problem instance is evaluated with the assignment and value of the evaluation (number of vertices in a clique, number of clauses satisfied, or number of vertices in a vertex-cover, and so forth) is scaled using the scaling parameter and eventually returned. The algorithm can be seen in Algorithm 1, and the geometric presentation of the combinatorial problem in Figure 3.

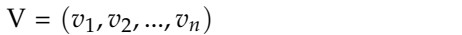

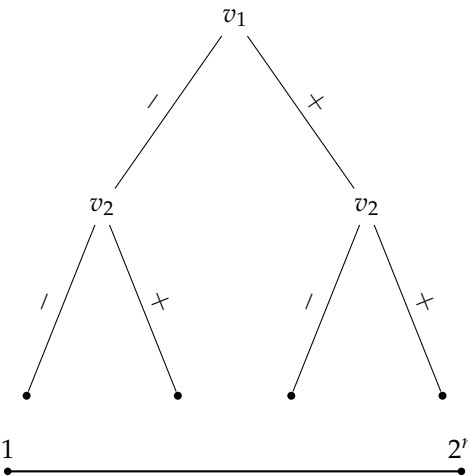

**Figure 3.** Geometric binary tree ($m = 2$) presentation of a combinatorial problem. Numbers on the bottom imply the domain.

If Algorithm 1 is called using a partial assignment (that is, some variables have already been assigned values), then the algorithm starts by randomly ordering the variables that have not been assigned to any values. Then, the algorithm produces the function for the remaining variables and uses the partial assignment as a basis of the final evaluation.

---

**Algorithm 1** Combinatorial problem to finite domain univariate function

1: **input: X (problem instance), V (variables), scale (scale of function values), P (partial assignment),** $D_0$**,** $D_1$
2: randomly sort V (if P contains values then discard assigned variables from V)
3: create black-box function with parameters x (point to evaluate), $[D_0, D_1]$:
4: current := $[D_0, D_1]$
5: $i := 0$
6: **while** current $\neq$ single value **do**
7:    a := first *half* of current
8:    b := second *half* of current
9:    current := a **if** x $\in$ a **else** b
10:   $V_i$ := {1 or 0} based on current (whether a or b was selected)
11:   $i := i + 1$
12: **end while**
13: assignment := evaluate X with assignment of V and P
14: y := value of assignment (value of maximization / minimization problem)
15: scale y based on *scale*
16: **return** black box function (lines 3–15)

---

In a combinatorial problem, whose combinatorial complexity is $O(2^n)$, there exist $2^n$ different variable assignments. Algorithm 1 runs $O(n)$ steps and at each step, the algorithm splits the remaining domain into two halves and selects another half as a new domain. The assignment for the combinatorial problem is updated based on the order of the variables, the depth at which the algorithm currently operates, and whether the value of input $x$ belongs to the first or second half of the split interval (other half indicates, say, 0 value and the other half indicates 1 value). In this manner, the algorithm assigns each value of input $x \in D, D = [D_0, D_1]$ ($[1, 2^n]$, for instance) with a different assignment, which is used to evaluate the combinatorial problem at the end of the algorithm. Then, the scaling parameter is used to scale the returned value if the new scale is required—in order to produce functions from a specific Gaussian process prior. Further, the scaling factor

can also be used to group certain optimization results for the same function values. For example, 1 or 2 clauses satisfied by an assignment yields a function value of 1.

The random ordering of the variables from the same combinatorial optimization problem instance always produces the same function values that lie in the different positions in the function domain—even if some variables are fixed as in the partial assignment's case.

*Example Run of Algorithm 1*

For the example run of Algorithm 1, we define a *maximum satisfiability* (MAX SAT) [33] problem with 3 boolean variables $\{x, y, z\}$. Let the set of clauses in the problem be $\{(x \lor y \lor z), (x \lor \neg y \lor z)\}$, where $\lor$ is a logical disjunction, and $\neg$ is a logical complement. Clearly, the instance can be satisfied through multiple assignments. A geometric presentation of the problem as a decision tree can be seen in Figure 4.

For the sake of an example, we generated one function without a partial assignment and two functions with a partial assignment to demonstrate Algorithm 1. Without a loss of generality, the functions and the problem instance are kept quite simplistic.

For the first example, let us assume that Algorithm 1 sorts the variables $\{x, y, z\}$ to order $(x, y, z)$. The domain of the function is clearly all natural numbers in the range $[1, 8]$ ($2^n, [n = 3] \rightarrow 8$). Let us then assume that from the geometric perspective, all intervals on the left from the two sibling intervals are 0 assignments and all intervals on the right are 1 assignments, and that the black box function from Algorithm 1 is called with a parameter of 3.

First, the function splits the domain $[1, 8]$ into two halves (lines 7 and 8 in Algorithm 1) $[1, 4], [5, 8]$ and checks whether 3 belongs to $[1, 4]$ or $[5, 8]$. Here, it is clear that $3 \in [1, 4]$. Because 3 belongs to the left interval $[1, 4]$, the first variable, which is $x$, is assigned with 0.

Second, the interval $[1, 4]$ is split into $[1, 2]$ and $[3, 4]$. This time, $3 \in [3, 4]$, which is the right interval. The second variable $y$ is thus assigned with 1. Finally, the intervals for the third variable $z$ are $[3]$ and $[4]$ and since $3 \in [3]$, the third variable, is assigned with 0.

The produced assignment for the parameter 3 is $(x = 0, y = 1, z = 0)$, which satisfies the clause $(x \lor y \lor z)$ but not the clause $(x \lor \neg y \lor z)$. Hence, the value returned by the function is 1 without any scaling.

In the second example, we show how sorting variables randomly after a partial assignment produces two different functions from the same problem instance. In the second example, we make a similar assumption that the left interval from two sibling intervals corresponds to the assignment of 0 to a variable and the right interval corresponds to the assignment of 1.

This time, Algorithm 1 is called with a partial assignment, $x = 0$. Thus, the remaining interval for the order $(x = 0, y, z)$ would be $[1, 4]$ because $x = 0$, which implies it is in the left interval of siblings $[1, 4]$ and $[5, 8]$. Figures 5 and 6 illustrate how the produced function changes if the variables $(y, z)$ are sorted differently for our problem instance. In Figure 5, the order is $(z, y)$ and the new parameter, this time, 2 would evaluate the assignment $(x = 0, z = 0, y = 1)$, which satisfies only one clause. However, if the order would be $(y, z)$ as in Figure 6, parameter 2 would yield the assignment $(x = 0, y = 0, z = 1)$, which satisfies both of the clauses. This demonstrates that functions from the same problem instance can output different values when the order of the variables is chosen randomly.

From these two examples, it is easy to see how combinatorial optimization problems are reduced to univariate functions and how different functions can be generated through randomness from the same problem instance.

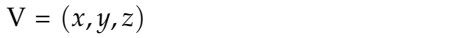

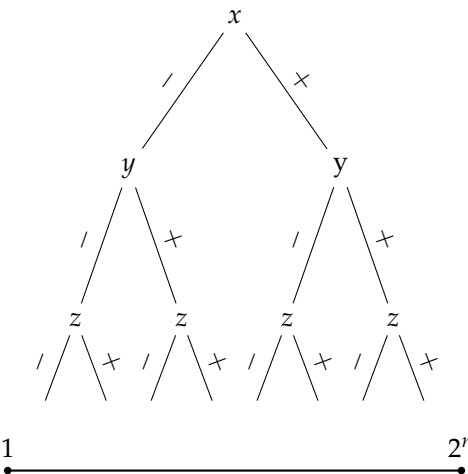

**Figure 4.** Decision tree presentation of MAX SAT with 3 variables. Numbers on the bottom imply the possible parameters for the univariate function.

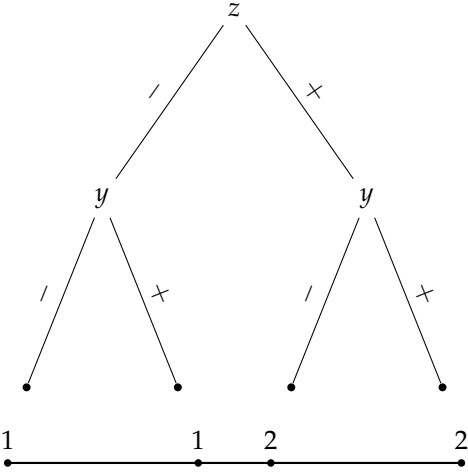

**Figure 5.** Geometric presentation of the MAX SAT problem with 3 variables where the order of the variables is $(x = 0, z, y)$ (The numbers on the bottom imply the clauses are satisfied by the assignment).

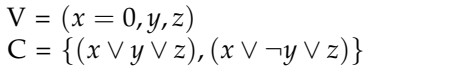

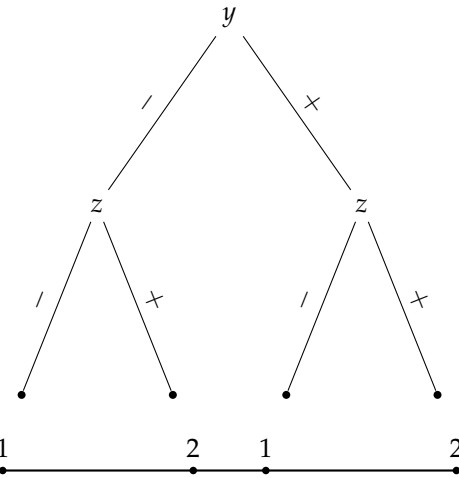

**Figure 6.** Geometric presentation of the MAX SAT problem with 3 variables, where the order of the variables is $(x = 0, y, z)$ (The numbers on the bottom imply the clauses are satisfied by the assignment)

## 4. Human–Algorithm Collaboration in Bayesian Optimization

In this section, we refer to a recent paper [27], where the authors showed that any ratio between expected global optimum and expected optimum found by their Bayesian optimization (BO) algorithms (UCB2 or EI2) can be found in the functions with finite domains in exponential time. We complement their results to show that for a specific type of problems, their results can be extended to a polynomial time human–algorithm collaboration procedure. Here, we stress that the algorithm is merely theoretical and is required to query Gaussian priors multiple times from an external expert.

The key ingredient of their work that we use is that instead of the standard simple regret used in BO they provided proof for the expected approximation ratio—the multiplicative ratio between the expected best value found by the function and expected global optimum (normregret in Equation (5)). Also, their bound of the regret ratio is tight for an arbitrary constant. That is, their UCB2 or EI2 are optimal to the worst case bound. This means that the upper bound equals the lower bound in their algorithms up to some constant $\epsilon$. Furthermore, their regret ratio is not asymptotic—which is crucial in our analysis. The upper and lower bounds, respectively, for their algorithms are

$$normregret \leq 1 - (1 - T^{\frac{1}{2\pi}}) \cdot \frac{\sqrt{\log_2(T) - \log_2(3\log_2^{\frac{3}{2}}(T))}}{\sqrt{\log_2(N)}} \tag{6}$$

and

$$normregret \geq 1 - \frac{\sqrt{\log_2(T)}}{\sqrt{\log_2(N)}} - \epsilon \tag{7}$$

where $T$ is the number of function evaluations, $N$ is the domain size, and $\epsilon$ is any constant with $\epsilon > 0$. The pseudocode of their UCB2 algorithm, which is quite similar to the ordinary UCB algorithm can be seen in Algorithm 2. The main difference between UCB2 and a standard UCB is that it is symmetrical: flipping the sign of the function has no any effect on the found values: instead, UCB2 optimizes both the highest and lowest function values.

$$x_t = argmax\ max\{-Y_t + M_t^n + \sqrt{C_t^{nn} \cdot 2 \cdot logN}, Y_t - M_t^n + \sqrt{C_t^{nn} \cdot 2 \cdot logN}\} \tag{8}$$

where $M$ is the posterior mean, $C$ is the given covariance,

$$C_t := COV(F|A_{:t}, T_{:T}), \tag{9}$$

and A is the domain $A := [1, D]$.

As such, their work cannot be extended to polynomial time algorithms to achieve a constant factor approximation in polynomial time due to the fact that their bounds are only *expectations* (as in standard BO). Hence, to obtain the strict ratio provided by their algorithm, one would have to evaluate the arbitrary number of different functions. We overcome this constraint by narrowing down the scope of the functions to the type of functions, where the function can be changed almost arbitrarily many times, but still depends on the same problem instance (see Algorithm 1).

---

**Algorithm 2** UCB2 algorithm.

---

1: **inputs: D (domain),** $\mu$**,** $\Sigma$**,** $T$ **(number of iterations)**
2: **for** $t := 0, 1, 2, ...$ *to* $T$ **do**
3:     $x_t := \text{argmax max} \left\{ -Y_t + M_t^n + \sqrt{C_t^{nn} \cdot 2 \cdot \log_2(N)}, Y_t - M_t^n + \sqrt{C_t^{nn} \cdot 2 \cdot \log_2(N)} \right\}$
4:     sample $y_t := F(x_t)$
5:     perform Bayesian update on posterior
6: **end for**

---

Without a loss of generality, we assume, once again, that the combinatorial problems here have a combinatorial complexity of $O(2^n)$. In Algorithm 1, we showed that any combinatorial problem with a combinatorial complexity of $O(2^n)$, can be reduced to a black-box univariate function with a finite domain. By running the Algorithm 1 multiple times for a single instance with different scaling factors, it produces different functions from the same problem instance. This is because the algorithm always randomly sorts either all or a subset of the variables (line 2 in Algorithm 1). We use that and *McDiarmid's inequality* [34] to obtain a high probability of a convergence to expected values. The constraint defined of a McDiarmid's inequality is

$$|F(x_1, x_2, ..., x_m) - F(x_1, x_2, ..., x_i', ..., x_m)| \leq c_i \tag{10}$$

where a value change of a single random variable can cause a function value to change to $c_i$ at most, which resembles a local Lipschitz condition.

Meanwhile, the actual concentration inequality is

$$P[|F(x_1, x_2, ..., x_m) - E[F(x_1, x_2, ..., x_m)]| \geq t] \leq exp(\frac{-2t^2}{\Sigma_{i=0}^m (c_i^2)}) \tag{11}$$

where $t$ is some constant for the difference between the function of random variables and expected value of the function with random variables. Moreover, McDiarmid's inequality is a concentration inequality that states that if a arbitrary function accepts $m$ random variables as parameters and a value of one those random variables is changed from the original value, then the function output can only change up to some constant from its original value (Equation (10)). Equation (11) is the actual concentration inequality that states that if Equation (10) holds for any function accepting $m$ random variables as parameters, then such a function's output converges exponentially fast to its expected value. Meanwhile, Hoeffding's inequality [35] is a special case of McDiarmid's inequality, where the function of random variables is their sum.

In our case, we run our reduction algorithm (Algorithm 1) $S$ times, use either UCB2 or EI2 [27] each time and finally output the mean of the found values by UCB2 or EI2. Based on McDiarmid's inequality, the mean value converges exponentially fast to the bounds promised by UCB2 or EI2. We then branch to the optimal regions of the domain where the optimal function values are likely to reside. Hence, our algorithm follows rather popular geometric branch and bound framework used in multiple global optimization algorithms [32].

In [32], the search space is shrunk after function evaluations and the intervals where the global optimizer is not likely to reside are discarded. However, ref. [32] cannot be used as such to solve problems with exponential sized domains because constants in their algorithm depend on the domain size and the shape of the function near to optimizer(s).

Because the prior given to the Gaussian process upper bounds the expected optimum and the best value found by UCB2 and EI2, at every expansion and new sample from Algorithm 1, we query a new prior from an external expert. Here, we assume that the priors queried from the expert do not contradict each others and are informative—that is, there are no conflicting choices of, say, hyperparameters, among the priors regarding the same fractions of the search space, and that the expert can gain insights on the functions.

Querying informative priors from an expert is a realistic assumption—and usually made in BO literature—since the expert has access to multiple sampling heuristics, such as maximum likelihood estimation, previous function evaluations, and *domain knowledge*. This differentiates querying priors from an expert from standard *oracle* queries from the theory of computation, where the oracle is unrealistic non-deterministic entity who is always correct. Our extension to [27] algorithm can be seen in Algorithm 3.

Before going into more detail, we will first introduce some definitions and an assumption.

**Definition 1** (Cell). *Interval produced when a domain or a part of the domain is divided into two halves: one division produces two new cells.*

**Definition 2** (Expansion of cell). *Division of a cell into two intervals of equal size.*

**Definition 3** (Upper bound). *Expected maximum of a cell. Upper bound is calculated from the size of a cell, mean maximum found using EI2 or UCB2 from samples of Algorithm 1, and the number of iterations used by EI2 or UCB2 (X):*

$$ub := \frac{val}{\frac{\sqrt{\log_2(T)}}{\sqrt{\log_2(N)}}} \tag{12}$$

*where val is the mean of values found by EI2 or UCB2 in S runs from a cell, $T = X$, and N is the size of the cell. Ub is calculated for each cell as if there were no approximation error.*

**Definition 4** ($\epsilon$ optimal solution). *An expected local optimum $\epsilon$ close to an expected global optimum.*

**Definition 5** (Optimal cell). *Cell that contains the $\epsilon$ optimal solution.*

**Assumption 1.** *The expert's priors do not contradict each others at any point in the same fractions of the search space, and priors are informative.*

In Algorithm 3 we show the pseudocode of the algorithm. If the priors given by the expert are informative and consistent, then it converges on the $\epsilon$ optimal solution in $O(S \cdot A \cdot K \cdot C \cdot \log_2(D))$; for some C, depending on the problem instance, *S* and *X* as well as *K* (the number of function values $\epsilon$ close to global optima). This is because Algorithm 3 with high probability expands the $\epsilon$ optimal cell into two equal sized halves—the rate of convergence is exponential. In the following lemmas, we will prove our claims.

---

**Algorithm 3** Bayesian optimization with expert knowledge.

---

1: **inputs: D (domain)**, *T*, *S*, *X*, *V*
2: current := $[1, D]$
3: $i := 0$
4: $j := 0$
5: fix order of *Vars* := *V*
6: **for** $t := 1, 2, 3, \dots$ *to* $\log_2(T)$ **do**
7:  a := first *half* of current
8:  b := second *half* of current
9:  remove current cell
10:  add a to cells
11:  add b to cells
12:  **for** $s := 1, 2, 3, \dots$ *to* $\log_2(S)$ **do**
13:   re-sample a and b from Algorithm 1, use absolute position of a and b (in original domain) to derive a partial assignment for Algorithm 1 and $D_0, D_1$.
14:   derive covariance matrices $(\Sigma_i, \Sigma_j)$, and means $(\mu_i, \mu_j)$ for a and b using query to **expert**
15:   solve UCB2 or EI2 for a, use $T := X$ iterations, covariance matrix derived $\Sigma_i$, and $\mu_i$
16:   solve UCB2 or EI2 for b, use $T := X$ iterations, covariance matrix derived $\Sigma_j$, and $\mu_j$
17:   $i := i + 1$
18:   $j := j + 1$
19:  **end for**
20:  retain original order of variables for *a* from *Vars*
21:  retain original order of variables for *b* from *Vars*
22:  deduce *ub* of *a* from its found UCB2 or EI2 values in S samples from Algorithm 1, size of *a*, and X
23:  deduce *ub* of *b* from its found UCB2 or EI2 values in S samples from Algorithm 1, size of *b*, and X
24:  current := *argmax* of cells (the cell with highest *ub*)
25: **end for**
26: **return** max value of any cell found by UCB2 or EI2

---

**Lemma 1.** *With probability in S, Algorithm 3 expands the optimal cell at any time $t_i \in T, 0 \leq i \leq T$*

**Proof.** Based on McDiarmid's inequality,

$$|F(x_1, x_2, \dots, x_m) - F(x_1, x_2, \dots, x'_i, \dots, x_m)| \leq c_i \tag{13}$$

and

$$P[|F(x_1, x_2, \dots, x_m) - E[F(x_1, x_2, \dots, x_m)]| \geq t] \leq exp(\frac{-2t^2}{\Sigma_{i=0}^m (c_i^2)}) \tag{14}$$

the Algorithm 3 can be modified so that it passes all the *S* random permutations from Algorithm 1 to a black box function that optimizes permutations with EI2 or UCB2 and outputs the mean of the results. Hence, $c_i$ is bounded by $\frac{1}{m}(b - a)$, for every *c*, where *b* and *a* are the upper bound and lower bound of the functions values, respectively. Consequently, we have the following:

$$P[|F(x_1, x_2, \dots, x_m) - E[F(x_1, x_2, \dots, x_m)]| \geq t] \leq exp(\frac{-2m^2 t^2}{(b - a)^2}) \tag{15}$$

This implies that the value of *S* UCB2 or EI2 runs converges exponentially fast to its expected value in *m*—the number of random permutations.

The expected approximation ratio in [27] is given as

$$1 - (1 - T^{\frac{1}{2\pi}}) \cdot \frac{\sqrt{\log_2(T) - \log_2(3\log_2^{\frac{3}{2}}(T))}}{\sqrt{\log_2(N)}} \tag{16}$$

which is

$$\Omega((1 - T^{\frac{1}{2\pi}}) \cdot \frac{\sqrt{\log_2(T) - \log_2(3\log_2^{\frac{3}{2}}(T))}}{\sqrt{\log_2(N)}}) \tag{17}$$

This is upper bounded by

$$o(\frac{\sqrt{\log_2(T)}}{\sqrt{\log_2(N)}}) \tag{18}$$

which is because bound in Equation (7) is for every constant $\epsilon > 0$. Based on McDiarmid's inequality, the mean approximation ratio tends to be in between these bounds in $m := S$. Note that these upper and lower bounds are not asymptotic.

The cell with the highest upper bound is always expanded. Thus, the upper, and lower bounds imply that the cell that does not contain the expected global optimizer might be expanded instead of the cell with the optimizer—some cells might have lower approximation errors and only slightly smaller expected optimums, and hence, there is a larger *ub*.

This is limited by

$$E[F_{sup}] \cdot (1 - T^{\frac{1}{2\pi}}) \cdot \frac{\sqrt{\log_2(T) - \log_2(3\log_2^{\frac{3}{2}}(T))}}{\sqrt{\log_2(N)}} > E[F_{local}] \cdot \frac{\sqrt{\log_2(T)}}{\sqrt{\log_2(N)}} \tag{19}$$

which is

$$E[F_{sup}] > E[F_{local}] \cdot \frac{\sqrt[2\pi]{T} \cdot \sqrt{\log_2(T)}}{(\sqrt[2\pi]{T} - 1)\sqrt{\log_2(T) - \log_2(3\log_2^{\frac{3}{2}}(T))}} + \epsilon \tag{20}$$

where $F_{local}$ is a maximum of a cell other than the expected global optima, and $\epsilon$ is a small constant from McDiarmid's inequality's convergence error. Equation (20) gives a limit for the $\epsilon$ optimal solution.

This implies that even for a cell without any approximation error, if the found value is less than (20) the factor from the expected global maximum, then the cell will not be expanded because it will be *dominated* by at least the cell with the expected global optimizer.

These prove that—with probability in $m := S$—only optimal cells will be expanded. □

**Lemma 2.** *Algorithm 3 will, with very high probability, find at least the $\epsilon$ optimal solution to any optimization problem derived from some Gaussian process and Algorithm 1.*

**Proof.** When running Algorithm 3 for a sufficient number of iterations, the values found by UCB2 and EI2 increase, as cells' size decrease [27], and tend the to cell's expected optimum.

The probability $x$ of *always* selecting the optimal cell is propositional to McDiarmid's inequality, $m := S$, but decreases exponentially based on the depth of the tree and tends to 0:

$$P[x|h] := (1 - (e^{-\frac{2m^2t^2}{(b-a)^2}}))^h \tag{21}$$

where $h$ is the maximum depth of the tree. On the other hand, because

$$P[x] := 1 - (e^{-\frac{2m^2t^2}{(b-a)^2}}) \tag{22}$$

$P[x]$ increases exponentially in $m$ and tends to 1; the expected required samples from Algorithm 1 to obtain $P[x|h] > 0.5$ is bounded polynomially in $m := O(log_2(D)) = O(h)$. $\square$

**Lemma 3.** *Expected run time of Algorithm 3 to find at least the $\epsilon$ optimal solution is $O(S \cdot A \cdot K \cdot C \cdot log_2(D))$ (the time complexity of UCB2 or EI2 is omitted here) if the function has K number of $\epsilon$ optimal solutions, where D is the domain size, A is the number of variables in the combinatorial optimization problem, and C depends on the function values and X.*

**Proof.** By Lemma 4.3, the algorithm always expands the optimal cell with high probability in $S$. Because the expansion procedure always divides the cell into two equally sized halves and because the $\epsilon$ optimal cell is expanded with high probability, the convergence rate is $O(2^{-n})$. Upon fixing $N = log_2(D)$, we have $O(S \cdot A \cdot K \cdot C \cdot log_2(D))$. $C$ depends on when Algorithm 3 is able to distinguish the expected global optimum from similar values of the objective function. This further depends on parameter $X$ for UCB2 or EI2. This completes our proof. $\square$

Of course, the approach provided here is merely theoretical. Here, the first assumption is that all functions are indeed realizations of the Gaussian processes (which might not be the case). The second is that, as the domain size increases, the more priors (even in the best possible case) the expert has to provide. Such an actions are not feasible even for moderate sized problems in real life. Moreover, a disadvantage of the algorithm is that some covariance functions, UCB2 or EI2, may produce better results and have less strict bounds because the bounds are general and not prior dependent. This, of course, leads to a situation where for some priors, Algorithm 3 spends more time in searching for sub-optimal regions of the search space.

A sample run of the search space shrinking can be seen in Figure 7. Here, Figure 7, the Algorithm 3 first splits the domain into two equally sized halves $(-v_1)$ and $(+v_1)$ and samples $S$ functions from the Algorithm 1 with partial assignments. Then, for both intervals, the UCB2 or EI2 algorithm is run $S$ times. The *ub* value of $(-v_1)$ is higher than that of $(+v_1)$. Hence, $(-v_1)$ will be split into $(-v_1, -v_2)$ and $(-v_1, +v_2)$ and $S$ samples of functions with partial assignments will be generated by Algorithm 1 for UCB2 or EI2. From the intervals $(+v_2)$, $(-v_1, -v_2)$ and $(-v_1, +v_2)$, the interval $(-v_1, +v_2)$ has the highest *ub* value. This implies that it will be split into two halves producing the leaves $(-v_1, +v_2, -v_3)$ and $(-v_1, +v_2, +v_3)$. Every time a smaller intervals is generated, the found values will increase by the definition of normregret.

The term $\epsilon$ is small. For instance, fixing $T := X = 10^7$ yields less than a 2% approximation error for large domains. In Equation (20), we showed that the approximation error is propositional only to $X$. In the next section, we highlight the relevant numerical tests to show how important the correct prior is for the approximation accuracy.

*Numerical Experiments*

We used the algorithms defined in Algorithms 1 and 3, with 50 randomly generated *maximum exactly-3satisfiability* (MAX E3SAT) [36] instances and 25 randomly generated *maximum satisfiability* (MAX SAT) [33] instances. MAX E3SAT is a generalization of a *satisfiability* problems where the goal is to find an assignment to satisfy the maximum number of clauses consisting exactly of 3 literals. In [37], it was shown that MAX E3SAT can be approximated within a factor of at least 8/9, if the instances are randomly generated and in the 7/8 factor for all instances unless $P = NP$ [36]. MAX E3SAT differs from MAX SAT in that the number of literals in a clause has no restrictions in the latter, which can be approximated within the factor 3/4 [33].

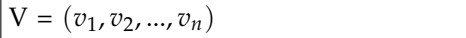

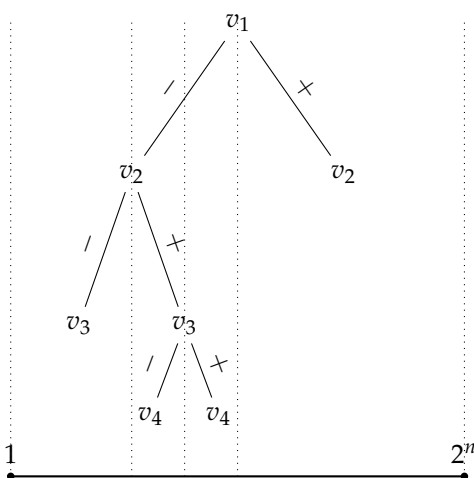

**Figure 7.** Algorithm 3 expands the cell with the highest upper bound $(-v_1, +v_2)$ and evaluates the upper bounds of cells $(-v_1, +v_2, +v_3)$ and $(-v_1, +v_2, -v_3)$ (upper bounds of cells $(+v_1)$, and $(-v_1, -v_2)$ are known).

For the numerical tests, we selected a discrete Wiener process as a statistical model for UCB2 because it does not require a hyperparameter selection during the iterations of Algorithm 3 if the variance is fixed as opposed to, say, squared exponential covariance (the lengthscale parameter is required, which may vary accross the domain).

A discrete Wiener process $W$ is a Gaussian process defined by Gaussian increments

$$W_{t+u} - W_t \sim N(0, u) \tag{23}$$

and

$$W_0 = 0 \tag{24}$$

The covariance function is defined in $W$ for $W_s$ and $W_t$, $t \geq s$ as

$$\min\{s, t\} \tag{25}$$

if the variance is fixed unit variance.

We selected MAX SAT and MAX E3SAT problems for numerical tests because they are well studied and understood optimization problems, and the function values are mostly non-zero everywhere (an assigment usually satisfies more than 0 clauses), as opposed to say, *maximum clique*. Hence, our hypothesis was that a discrete Wiener process would be a suitable statistical model for Algorithms 2 and 3.

For the tests, we bootstrapped UCB2 by randomly selecting $C$ points from the domain and used $T := C$, $S := T$, where $C$ was the number of clauses. The instances in MAX E3SAT had 125–350 clauses and in MAX SAT case, the instances had 125–250 clauses. We restricted the number of clauses because the standard implementation of UCB2 (or EI2) has a time complexity of $O(N^3)$, which yields a total of $O(N^5)$.

We evaluated the results by calculating the number of satisfied clauses and comparing the number to the overall number of clauses in the instance. Note that some instances were not satisfiable with high probability. For the MAX E3SAT, the highest achieved result was 0.944 of clauses being satisfied, while the lowest result was 0.792. The mean value of all 50 instances was 0.91 after rounding.

For the MAX SAT, where the number of literals in a clause was not restricted and varied from 1 to 5 in every instance, the lowest percentage of satisfied clauses was 0.63, while the highest percentage was 1.0 of clauses satisfied. On average, the number of clauses satisfied was 0.77 after rounding.

The results from the numerical experiments indicate that the methods proposed in this paper offer quite powerful approximation methods for NP-hard optimization problems also in practise. However, it must be noted that the instances we used (random MAX SAT and MAX E3SAT) were actually not quite the realizations of the discrete unit Wiener process—which might have caused the low approximation ratios for some of the used instances. UCB2 (and EI2) and Algorithm 3 require the function to be a realization of a given Gaussian process to provide a high approximation accuracy described in the previous section. The results can be seen in Table 1.

**Table 1.** Results from numerical tests . Note that OPT may be less than 100 since some instances were unsatisfiable with high probability.

| Optimization Problem | Instance Sizes (Clauses) | Number of Instances | Min Value % | Max Value % | Mean Value % | Inapproximability Bound |
|---|---|---|---|---|---|---|
| MAX E3SAT | 125–350 | 50 | 79 | 94 | 91 | 87.5 |
| MAX SAT | 125–250 | 25 | 63 | 100 | 77 | 75 |

In the future, we wish to extend the numerical studies to the problems that are usually considered harder to approximate than MAX E3SAT (or even MAX SAT). Such problems include, for instance, *maximum clique*, *minimum satisfiability*, *minimum vertex cover*, and *maximum DNF-satisfiability*. However, a Wiener process might not be a suitable statistical model for them because in some cases, the functions would be zero in most parts of the domain. Despite using only MAX E3SAT and MAX SAT instances and problems that were only of moderate size, the results are quite encouraging.

## 5. Results and Discussion

In this paper, we showed how a human–algorithm collaboration can lead to approximate combinatorial optimization problems with arbitrary high accuracy. We further showed that the theoretical guarantees of the method are at least partly supported with numerical experiments under the Wiener measure.

Our method casts new light on how NP-Hard optimization problems could be approximated in the future. Many of the problems are usually considered intractable and almost surely impossible to approximate accurately in a feasible time.

Through the proposed method, two possible approximation methods exist: first, an external expert can collaborate with an optimization algorithm and provide Gaussian priors with a proper set of hyperparameters. Second, one can rely on statistical models without any hyperparameters (such as the Wiener process) and use the methods introduced without any interaction.

The first method is more flexible and captures the full scope of the different Gaussian processes whereas the latter method restricts the optimization process by relaying only to a few possible statistical models. We saw in our numerical tests that for some instances, the Wiener process is not a suitable model for MAX SAT or MAX E3SAT. On the other hand, the second method does not require expertise from the user to have knowledge on the prior or as a matter of fact, any interaction between the algorithms and user.

While the proposed methods can be used to optimize problems in NPO, it should be noted that the scope of the solvable problems is not restricted to only search problems with a combinatorial complexity of $c^N$ for some constant $c$—all problems with less than double exponential combinatorial complexity can be used for at least sub-exponential time approximation. This might well include search problems outside of the polynomial hierarchy or PSPACE—the problems that are provable intractable (say, optimization problems in EXPTIME).

While the methods provide a powerful new alternative for approximating search problems, they could lead to a better understanding of the search problems as the expert places Gaussian priors over the search domain intervals as well. Here, placing a prior on a function requires understanding and knowledge of the structure of the problem. This

indicates that these methods could cast new light on the hidden structures of optimization problems or even help in answering one of the most fundamental questions in computing:

*Do NP optimization problems have any structure?*

However, it is important to note that more numerical tests should be conducted and the scope of problems in these tests should be expanded. In this paper, it was seen that placing a prior has a direct effect on the optimization accuracy. However, it was clear that, the priors in some problems that we used in the numerical tests were not suitable. Additionaly, the lower and upper bounds for the discrete Wiener process in UCB2 might not be as tight as they are for some other covariances. The tightness assumption of the bounds is a crucial ingredient in our proofs as it has a direct effect on the optimization accuracy.

To the best of our knowledge, we are the first to present a general purpose optimization method that, in theory, can be used to approximate any combinatorial optimization problem whose combinatorial complexity can be even super-exponential in arbitrary high accuracy. Even if the widely suspected conjecture $P \neq NP$ holds, the methods similar to this can be used to derive the feasible approximation results from intractable optimization problems.

## 6. Conclusions

In this paper, we have shown that with a type of HITL BO any combinatorial optimization problems (either minimization or maximization) can be solved in polynomial time as long as the combinatorial complexity of the problem is $O(poly(n)^{poly(n)})$ and we can assume that expert can correctly provide $poly(n)$ Gaussian process priors. While our proposed algorithm is merely theoretical and has only little practical interest, we assume that our approach could be used to gain new insights on how to tackle the most difficult NP-hard combinatorial problems in the future.

As it is usually expected, most of combinatorial optimization problems might be intractable. However, our research shed new light on how to avoid the common pitfalls faced while solving these problems: involving human expertise in the optimization process. In our approach, the human expert is not required to know where the optimizers lie in the search space. Instead, the expert is only assumed to provide a Gaussian prior when the expert is queried from. The prior captures expert's opinions on how drastically the nearby values might change in the function and whether the changes are periodic and so forth.

Additionally, the expert knowledge can be supported with previously calculated values of the objective function and, for instance, maximum likelihood estimation [32]. In this context, ref. [32] gave bounds on *sample sizes* of random variables (in our case, the function values) to derive a certain probability in the optimization of a likelihood function for covariance hyperparameters. Because the bounds do not depend on the domain size of the function, fixing a sample size and a covariance function for the covariance matrix $\Sigma$ yields a constant time estimation of a prior. This can be combined with the expert's knowledge.

Our results indicate that in the future, in order to understand the fundamental limits of our tools (whether it is a combinatorial optimization algorithm or an AI system) a human interference might be required. The scope of possible comibinatorial problems that can be solved efficiently in theory through our human–algorithm collaboration procedure is significant. Such problems include *protein structure prediction, finding maximum sized cliques, model checking, automated theorem proving*, and so on (basically every problem in NPO and even extending to problems with super-exponential combinatorial complexity).

**Author Contributions:** Conceptualization, T.T. and M.T.; methodology, T.T. and M.T.; software, T.T.; validation, T.T. and M.T.; formal analysis, T.T. and M.T.; investigation, T.T. and M.T.; resources, T.T. and M.T; data curation, T.T. and M.T.; writing—original draft preparation, T.T.; writing—review and editing, T.T. and M.T.; visualization, T.T.; supervision, M.T.; project administration, M.T.; funding acquisition, M.T. All authors have read and agreed to the published version of the manuscript.

**Funding:** This research received no external funding.

**Institutional Review Board Statement:** Not applicable.

**Informed Consent Statement:** Not applicable.

**Data Availability Statement:** For numerical tests, we used randomly modified instances from SATLIB (https://www.cs.ubc.ca/~hoos/SATLIB/benchm.html, accessed on 11 August 2021).

**Conflicts of Interest:** The authors declare no conflict of interest.

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
