# Peer review of "The Power of Human–Algorithm Collaboration in Solving Combinatorial Optimization Problems"

_algorithms, doi:10.3390/a14090253_

Round 1

Reviewer 1 Report

This paper has presented the power of human–algorithm collaboration in
solving combinatorial optimization problems. Generally, the paper is well written, but I suggest many points to improve the quality of the article.

1) I would also recommend the authors do more simulations to enhance their proposed method.

2) Some case studies can be presented to show the power of the proposed model.

3) The abstract has been briefly written and should be enriched by adding the main ideas and contributions.
4) The main contribution of the paper should be highlighted and emphasized. 

5) A separate section should be added for discussion of obtained results and main achievements.

6) Mention the constraints considered for optimizing equations 1 and 2.

7) Explain the equations (8) and (9) for better understanding to the readers.

Author Response

First, we would like to thank the anonymous reviewers for their constructive

critique and proposals. In this note, we discuss what changes we have made to

our manuscript after the proposals.

R1 stated that cases studies should be conducted in order to demonstrate the

power of the proposed methods |We conducted 2 cases studies with MAX SAT

and MAX E3SAT problems. We generated random instances of the problems

are compared the results to inapproximability bounds assuming that P 6= NP.

The instances of the problems were kept quite small because the Algorithms

have high (polynomial) time complexity. Despite this, the results were quite

encouraging. Although in some cases, the statistical models we selected did not

model the functions very well. R1 stated also that a new section should be added

to discuss the results and the achievements. We considered this quite important

proposal and wrote a new section where we discussed about the ndings and

implications of the research. The third proposal of R1 was that the abstract

should be re-written and should discuss the ndings and the results more. We

extended the abstract and now it gives clear picture of the implications of the

research. We also explained the inequalities better as R1 suggested and high-

lighted the achievements of the manuscript better. We furthermore mentioned

the constraints of the optimization problems in the manuscript as suggested by

R1. We also revised the language used in the manuscript.

Reviewer 2 Report

The author has shown that with human-algorithm interaction, the Bayesian Optimization algorithm can solve any combinatorial optimization problem in polynomial time assuming that the complexity of the problem is O(2^n) and the expert can correctly provide O(n) Gaussian process priors. Because of its assumptions and limitations, the proposed algorithm has rather limited practical applicability. However, the results presented in the paper are very interesting and important for the future research on human-algorithm (or human-AI) interactions.

The following issues should be addressed:

1. MDPI template should be used for paper formatting.

2. The English language should be improved – there are some grammar and style errors.

3. Figure 2 is poor quality – it is blurry.

Author Response

First, we would like to thank the anonymous reviewers for their constructive

critique and proposals. In this note, we discuss what changes we have made to

our manuscript after the proposals.

R2 suggested that one of the gures had poor quality. We replaced the gure.

We also corrected errors in the language as R2 suggested. R2 suggested that

we should use a dierent template. We note this suggestion but unfortunately

did not have enough time to do this. We will use the correct template if the

manuscript was accepted.

Reviewer 3 Report

This work reduces a class of combinatorial optimization problem instances to a univariate finite domain function, which can be approximated by human-algorithm collaboration Bayesian optimization algorithm.
Although the proposed algorithm is merely theoretical, the approach could be used to gain new insights on how to tackle the hardest NP-hard combinatorial problems.

Some more detailed comments are given below. I hope that if the authors will take them into account the paper will be improved.
1. It is hard to follow the algorithm 1 from combinatorial problem to finite domain univariate function. 
A simple example is needed to provide an illustration about algorithm 1 in section 3.
2. It is also hard to link the algorithm 2 with Bayesian optimization (BO) algorithms (UCB2 or EI2) in functions with finite domains.
A simple example of UCB2 or EI2 is needed to provide an illustration about algorithm 2 in section 4.
3. Remove the subsection title of 3.1 and 4.1, since there is only a subsection in section 3 and 4.
4. Please provide more than three keywords.
5. Number citations consecutively in square brackets.
6. Please ensure that every reference cited in the text is also present in the reference list (and vice versa). This article does not cite the Ref. [18], [27], [29], [33] in the text.

Author Response

First, we would like to thank the anonymous reviewers for their constructive

critique and proposals. In this note, we discuss what changes we have made to

our manuscript after the proposals.

R3 suggested that we should explain the Algorithms that we proposed better.

We added examples how the algorithms work and added a pseudocode for UCB2

algorithm. We also removed the sections that R3 proposed. We also revised the language in the manuscript.

We apology that we did not have enough time to revise the citations. We had to conduct several cases studies suggested by the another reviewer. We note that we will correct the citations if the manuscript is accepted.

Round 2

Reviewer 1 Report

All the comments are addressed. This paper may be accepted for publication.

Reviewer 3 Report

The authors have carefully addressed the previous comments of the reviewer and significantly improved the manuscript.